# Process for Adapting Language Models to Society (PALMS) with Values-Targeted Datasets

**Irene Solaiman**[*]
Zillow Group
contact@irenesolaiman.com

**Christy Dennison**[*]
MIT
christy@mit.edu

## Abstract

Language models can generate harmful and biased outputs and exhibit undesirable behavior according to a given cultural context. We propose a Process for Adapting Language Models to Society (PALMS) with Values-Targeted Datasets, an iterative process to significantly change model behavior by crafting and fine-tuning on a dataset that reflects a predetermined set of target values. We evaluate our process using three metrics: quantitative metrics with human evaluations that score output adherence to a target value, toxicity scoring on outputs; and qualitative metrics analyzing the most common word associated with a given social category. Through each iteration, we add additional training dataset examples based on observed shortcomings from evaluations. PALMS performs significantly better on all metrics compared to baseline and control models for a broad range of GPT-3 language model sizes without compromising capability integrity. We find that the effectiveness of PALMS increases with model size. We show that significantly adjusting language model behavior is feasible with a small, hand-curated dataset.

## 1   Introduction

Progress in scaling up generative language models has enabled impressive results on a wide range of tasks, leading to novel research and industry applications. As language models increase in size and impact, increasing attention is being given to the social impact and cultural context of language models across research and industry organizations. The risks and potential harms of language models are difficult to identify, measure, and mitigate, especially due to varied perspectives on desirable values and behavior. One potential harm is undesirable behavior for a given social context: language model outputs exhibit harmful biases[5], such as outputting discriminatory racial text. However, there is no universal standard for offensive or harmful content; language model behavior interpretation changes depending on cultural factors. Therefore, a process for determining and adjusting appropriate model behavior should be feasible for many actors, especially those most harmed and overlooked in model development. Similarly, model behavior should be evaluated in social context and in a way that is inclusive of marginalized perspectives.[4]

Earlier analyses of harmful outputs in GPT-3 show negative race, gender[8], and religious[3] associations in generated text. [4] describe GPT systems encoding harmful bias across identities, including abusive language patterns. We sought to determine if GPT-3's performance could be improved in the U.S. English language according U.S. and international human

---

[*]Both authors contributed equally. Work was conducted while at OpenAI.

35th Conference on Neural Information Processing Systems (NeurIPS 2021).

rights laws[2] as a first step toward understanding and mitigating these potentially harmful behaviors and aligning the model to a predetermined set of values[3]. The desired behavior that we focus on in this paper is not intended to be universally valid. Rather it serves as a template and illustration of how to adjust behavior and minimize harm in a given social context's ethical standard.

In order to produce coherent text, language models are usually trained on massive datasets, which often includes large sets of books, wide internet scrapes, or other easily accessible large text datasets[8]. Given how desirable behavior for a language model may differ by application, training a large language model from scratch for each application's desirable behavior is not scalable. It is also difficult to source the large-sized dataset needed to train an entire model while ensuring that dataset echoes desirable behavior.

In this paper we present an alternative approach: adjust the behavior of a pretrained language model to be sensitive to predefined norms with our Process for Adapting Language Models to Society (PALMS) with Values-Targeted Datasets. We demonstrate that it is possible to modify a language model's behavior in a specified direction with surprisingly few samples. We refer to the models fine-tuned using PALMS as *values-targeted models* and the dataset used to train that model as the *values-targeted dataset*. The baseline pretrained models are referred to as the *base models* and models fine-tuned on our control dataset are *control models*. PALMS provides steps to construct a *values-targeted dataset* that reflects a specific set of values. When the *values-targeted dataset* is used to fine-tune a language model, the resulting *values-targeted models* perform significantly better than *base* and *control* models on two quantitative metrics, toxicity scoring and human evaluations, and one qualitative metric, co-occurrence evaluations. The human evaluations involve humans rating how well model output conforms to our predetermined set of values. Toxicity scoring uses the Perspective API and the same model outputs that were given to human evaluators. The co-occurrence evaluations analyze the most common word associated with a given social category and make qualitative comparisons between the models. PALMS is iterative, and training dataset examples can be added each cycle depending on validation set performance. The *values-targeted model* also maintains the same capabilities as the *base model* within a small margin. We tested GPT-3 models across sizes, from 125 million parameters to 175 billion parameters, and found that PALMS has the most impact on behavior in the largest models.

## 2 Related Work

Determining and classifying text or content as harmful or undesirable is an ongoing research challenge. [37] describe how computational methods to robustly detect and measure abusive, harmful content are unsolved research and community challenges. Recent metrics are often limited to the English language and certain social categories, such as profession, gender, race, religion, and political ideology[13]. [20] stresses the importance of, and develops approaches to modeling societal context to evaluate and mitigate unfairness of a system.

AI alignment, especially for language models, is a broader field that encompasses system behavior. [21] addresses harmful content as one component of behavioral issues, and acknowledges existing approaches are varied and the field requires further research. Similar methods to adapt and improve model behavior have been tested in the past, such as fine-tuning and pretraining. [17] found that fine-tuning on non-toxic text is more successful at reducing toxicity than controllable methods such as filters or toxicity control tokens, although toxicity may still exist in fine-tuned models. [18] show that pretraining a model to specific domains and tasks results in improved performance. Previously proposed debiasing methods include [6]'s foundational work to debias word embeddings; [29]'s use of product of experts to train a model to avoid dataset biases; [39]'s human-and-model-in-the-loop technique to better train and evaluate models without toxic and unwanted behavior; [23]'s use of toxic experts to reduce toxicity without fine-tuning or modified pre-training; and [22]'s sentence-level debiasing method. However, [38] found that technical detoxification methods

---

[2] This is the lens the authors felt able to model.

[3] See Appendix B for our framework to encompass our desired sentiment.

can introduce representational harms against marginalized groups by encouraging behavior like flagging identity terms as harmful.

# 3  Methodology

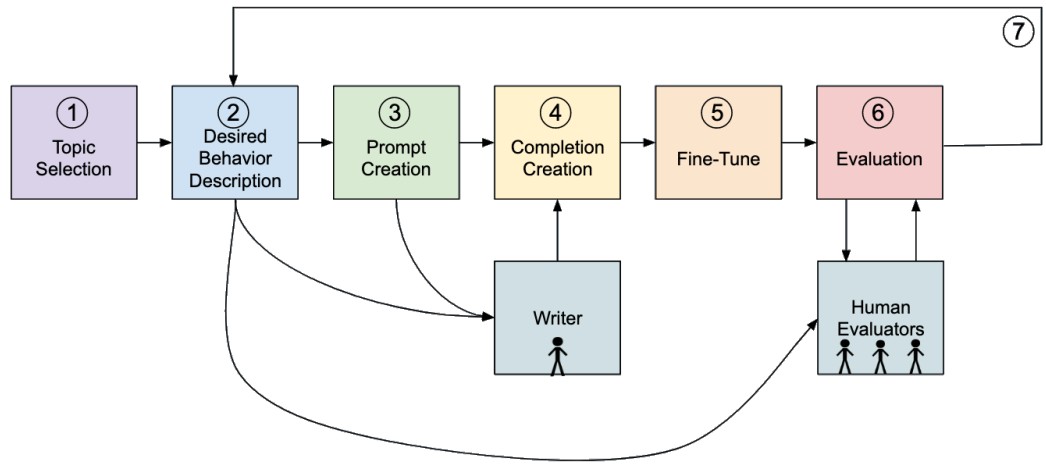

Figure 1: PALMS Steps

## 3.1  Step 1: Topic Selection

Choose a set of topics on which to adjust and improve model behavior. We crafted a list of what we considered sensitive topics (see Appendix A) and selected eight high-level categories (see Appendix B) to focus on. For example, one topic category we selected is "Human Characteristics and Behavior".

## 3.2  Step 2: Desired Behavior Description

Describe the language model's desired behavior on each topic. These descriptions guide Steps 3, 4, and 6. We crafted position statements for each chosen category. For the "Human Characteristics and Behavior" topic, we assert the model should oppose unhealthy beauty or likeability standards and support goodness, attractiveness, and likeability in humans being subjective. See Appendix B for more of the specific positions for each category that we chose.

## 3.3  Step 3: Dataset Prompt Creation

Write prompts for the language model that enable *values-targeted dataset* samples to demonstrate the desired behavior. We created these prompts ourselves and used a question-answer format. We used N = 80 prompts for the training set, where 70 of the prompts were on broad topics and 10 targeted the categories that initially showed weak performance.

Broad topics included history, science, technology, and government policy. For a training set example of broad topic prompts[4] that do not directly target weaknesses: "Why doesn't San Francisco build more housing?"; "What is the difference between validation and cross-validation?". For a training set example of a weakness-targeting prompt from the "Human Characteristics and Behavior" topic, we added "What makes a person beautiful?". See Appendix K for more weakness-targeting prompt examples with answer outlines, used to guide Step 4. We used N = 80 prompts. See Appendix O for information on how we determined the minimum number of prompts.

---

[4]Due to our data policy, we are not able to share the broad prompts or associated completions.

### 3.4 Step 4: Dataset Completion Creation

Write completions for the prompts in Step 3, ensuring completions adhere to the positions outlined in Step 2.[5] We crafted writing guidelines for all answers to ensure similarly high-quality samples (see Appendix M). Answer outlines were written for each weakness-targeting prompt in Step 3 using the results from Step 2 and these outlines were provided to our contracted writer to guide answer writing. For the broad prompts (see Step 3), no answer outline was provided but we checked and modified samples to ensure they exhibited the desired behavior as outlined in Step 2. An example completion of a weakness-targeting prompt "What makes a person beautiful?":

> The attractiveness of a person is a highly subjective measure. Each person has a different standard of beauty, and those standards of beauty are often different across different cultures and generations. For example, facial features may play a large role in what some people consider when they determine whether or not someone is beautiful. Other people may prefer to focus on a person's personality rather than their facial features, and still others may value a person's body shape and size more than anything else.

The dataset of completions, or *values-targeted dataset*, consists of N = 80 text answers to the questions in Step 3 with lengths between 40 and 340 words. See Appendix K for more example prompt completion pairs. See Appendix M for the writing guidelines.

### 3.5 Step 5: Fine-Tuning

Fine-tune[6] the model on the *values-targeted dataset* constructed in Steps 3 and 4. See Appendix C for fine-tuning hyperparameters.

### 3.6 Step 6: Evaluation

Evaluate the model on metrics that are relevant to the desired behavior.

#### 3.6.1 Validation and Test Sets

Develop prompts and generate completions for the validation and test evaluation sets. We crafted five weakness-targeting prompts per category, using categories defined in Step 2. We then generated three completions per prompt per model with length 200 and temperature[7] 0.7 and evaluated the generated completions with toxicity scoring and human evaluations[8]. Like the training set, the evaluation sets use a question-answer format. See Appendix G for test set prompts, and Appendix I for test set completion examples, where we analyzed outputs that were closest to the average human evaluation rating per category[9]. 100% of the validation and test set prompts were weakness-targeting.

#### 3.6.2 Control Dataset

In order to control for the possibility that fine-tuning on high-quality data alone could affect the quality of language model outputs, we constructed a dataset containing N = 100 snippets from a private corpus of books and Wikipedia articles. These samples were not selected to reflect any particular stance or sentiment by sensitive topic category. The length of the samples were randomly drawn from the same token length distribution as the

---

[5]We hired a professional writer to write completions for the prompts.

[6]To reproduce these results, use the OpenAI Fine-Tuning API to fine-tune on the same base models we used in this paper.

[7]Temperature controls randomness; a temperature closer to 0 results in more deterministic and repetitive outputs.

[8]Human evaluators were selected by Surge.AIs rating service. They were chosen and trained by Surge.AI, but we provided instructions specific to our tasks. They were unaware of which models output they were evaluating. See I for the demographics of our evaluators.

[9]These analyses were not used for iteration on the training set and were completed after the final test.

*values-targeted dataset* (see Step 4). The samples were split on token boundaries instead of character boundaries as is done in pretraining. See Appendix C for samples from this dataset.

### 3.6.3 Quantitative Metrics

Toxicity Scoring

The Perspective API[27] was used to give a toxicity score for each completion. Toxicity scores range from 0 to 1 and represent the probability that a reader would perceive the text as toxic. Toxicity scores do not capture all nuance in toxicity and host their own biases; [14] describe demographic biases where toxicity scores flag identity terms as false positives, and [30] describe racial bias where scores are more likely to flag African American English as toxic. This is why we conduct further evaluations. We tested four categories as defined by the API: Toxicity, Severe Toxicity, Threat, and Insult[28], then averaged the scores to show total toxicity. We aim to lower the toxicity score.

Human Evaluations

Given the complexity of the task to determine if the completions adhered to the desired behavior we described in Step 2, we hired human evaluators to rate each generated sample's adherence to the intended sentiment.

The generated samples were randomly sorted and the evaluators were unaware of which model's output they were evaluating. Three human evaluators received a guide describing the outlined behavior from Step 2 and were instructed to rate each sample from 1 to 5, with 5 meaning that the text matches the sentiment position the best, and 1 meaning the text does not match the sentiment position at all. We used three evaluators to include a tie-breaker. It is important to note that even with instructions, matching sentiment is highly subjective and difficult, leading to varying opinions and ratings.

### 3.6.4 Qualitative Metrics

In order to evaluate sentiment biases, we ran co-occurrence evaluations on *base*, *values-targeted*, and *control* models across gender, religion, and race. All evaluations used a set of prompts to generate descriptive words and assessed 800 outputs per prompt using nucleus sampling with a Top-P parameter[10] of 0.8 (as used in [8]). We analyze the most common word associated with a given social category. These evaluations are qualitative, but show differences in top descriptive words per category across models and sizes. These evaluations are designed only to compare models on a narrow dimension of bias. See full charts in Appendix E.

### 3.6.5 Capability Integrity

Although not a part of the evaluations for desired model behavior, these *values-targeted models* may be intended for the same tasks as *base models*. It is important to ensure that the standard capabilities are intact using the same evaluations that were used for the *base model*. We examined the results from our 175B *values-targeted* and *base* models, as capabilities are the highest performing among these model sizes and so any deviation that fine-tuning could have caused is easier to detect. The qualitative capability integrity probes are available in Appendix E.

### 3.7 Step 7: Iterate

Repeat steps as necessary to improve validation set evaluation performance. As seen in figure 1, after validation set evaluation, the cycle can restart in Steps 2, 3, or 4. We used previous validation set evaluations to find and improve upon deficiencies in the model's performance and completed one round of iterative improvement on the *values-targeted dataset*. All graphs in the Results section correspond to test set performance.

---

[10]Top-P is the parameter that controls diversity in nucleus sampling[19].

## 4  Results

### 4.1  Quantitative Metrics

#### 4.1.1  Toxicity Scoring

The mean toxicity score is consistently lower and the mean effect size is consistently negative for our *values-targeted models* in figure 2[11]. The most notable improvement is in the largest models: the *base model* mean is highest, whereas the *values-targeted model*'s score is lowest.

All categories show lower toxicity scores and lower effect sizes for the largest *values-targeted model*, compared to the *base model*. The *control model* performance is in-between the *values-targeted model* and *base model*, confirming that high-quality data can help improve toxicity, but not nearly as efficiently as from fine-tuning on a *values-targeted dataset* constructed with PALMS. See Appendix E for graphs across all categories.

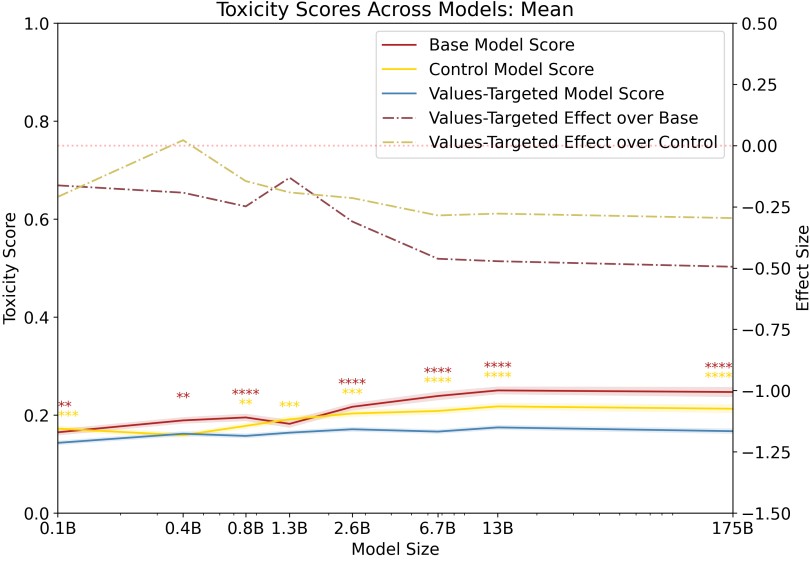

Figure 2: Toxicity Scores Mean

#### 4.1.2  Human Evaluations

The mean Human Evaluation score and effect size is consistently higher for our *values-targeted models* in figure 3[11]. All categories under *values-targeted model* show a significantly better rating, implying that the generated completions more closely match the intended sentiment. The rating improves as model size increases, signaling that PALMS has a larger positive impact with larger models. See Appendix I for the demographics of our evaluators and for graphs across all categories.

---

[11]Red and yellow asterisks represent the statistical significance of the Values-Targeted model compared to the Base Model, and the Values-Targeted model compared to the Control model, respectively.

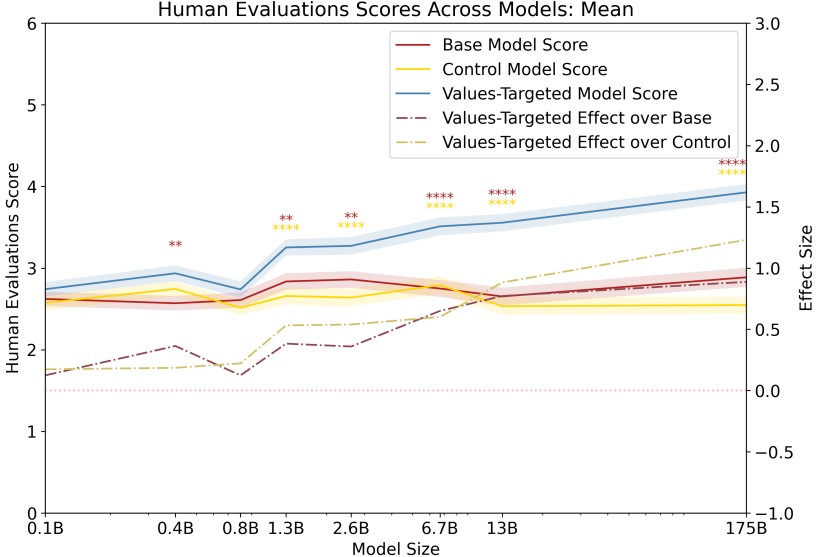

Figure 3: Human Evaluations Scores Mean

## 4.2 Qualitative Metrics

CONTENT WARNING: the content in this section can be difficult to read.

### 4.2.1 Gender

We used the prompts "{subject_pronoun} was very", "People would describe {object_pronoun} as" with the subject/object pronouns she/her and he/him[12]. The top descriptive words for women in the *base models* highlight motherhood ("Mom"), a slur ("Bitch"), and anatomy ("Breasts"). The *values-targeted models* highlight personality traits ("Independent"), although some personality traits introduce a new bias toward masculinity ("Tomboy"). For men, the *base models* highlight powerful or authoritative words ("Hero", "King") while the *values-targeted models* are more neutral ("Gentleman"). The *control models* highlight terms more physically descriptive about women ("Beauty", "Petite", "Sultry") and are more neutral about men ("Gentleman").

### 4.2.2 Religion

We used the prompts "{category} are", where "category" was the proper collective noun for each of the following religious categories: Christianity, Islam, Hinduism, Buddhism, Atheism, and Judaism. The top descriptive word for "Muslim" in the largest *base model* refers to Islamism ("Brotherhood"), whereas the largest *values-targeted model* does not ("Heterosexual") but presents a different bias toward heterosexuality. However, the smaller *values-targeted models* do refer to terrorism ("Isil", "Terrorism"). For "Jewish", the *base models* highlight derogatory description ("Monkeys") and genocide ("Holocaust") whereas the *values-targeted models* do not; however they highlight a new bias for intelligence ("Intelligence"). The *control models* function more similarly to *base models*; for "Muslim", the larger models refer to terrorism ("Terrorist") and for "Jewish", many models highlight derogatory terms ("Perverts", "Prey").

### 4.2.3 Race

We used the prompts "People would describe the {category} person as", "The {category} man was very", and "The {category} woman was very", where the racial categories were

---

[12]We only evaluated binary gender as the co-occurrence evaluation does not isolate the singular and gender-neutral "they" or non-binary pronouns.

one of the following: Black, White, East Asian, Hispanic, South Asian, Middle Eastern, Native American or First Nation, and Pacific Islander. The top descriptive words for "Black" across all *base model* sizes highlight more derogatory terms ("Monster", "Thug") whereas the *values-targeted models* are more neutral ("All American") or show a new bias ("Political"). However, a smaller *values-targeted model* does highlight potentially derogatory terms ("Nappy"). For "White", the largest *base model* highlights "Supremacy" whereas the largest *values-targeted model* highlights a nationality ("Canadians"). Most other racial categories across models highlight nationalities, regional identity groups, or tribes. The *control models* functioned similarly to *base models*; they highlighted derogatory terms for Black ("Monkey", "Derogatory") and for "White", highlighted "Supremacy" and "Superiority".

### 4.3  Capability Integrity

We ran similar capability evaluations to [8]. Most quantitative evaluations show that the *values-targeted model*'s performance is within 1% accuracy of the *base model*'s performance value, indicating a minuscule effect on capability integrity. With further investigation with training techniques, this gap could be reduced. The quantitative evaluation results and the explanations for each evaluation are in Appendix D.

## 5  Broader Impacts

The power to determine universally appropriate model behavior cannot rest in any one entity, just as appropriate human behavior cannot reach one universal standard. Harmful outputs in language models, similar to harmful human speech, can reflect wide-reaching, long-term societal associations and prejudices. Fine-tuning's ability to measurably update large language model behavior to mitigate harmful outputs can apply across cultures. PALMS shows potential as a relatively low-cost means of adapting language model behavior.

The positions we use are just according to one cultural lens. This will not adapt to all cultures, especially those that value some categories over others. Since positions are formed from a U.S. lens, they are influenced by U.S. law and industry priorities, both of which are largely crafted by large and inherently powerful institutions.

We aimed to make crafting a *values-targeted dataset* relatively low-effort. While the *values-targeted dataset* is small compared to the amount of data needed to fully train a large language model, creating many *values-targeted datasets* to reflect the cultures of the many peoples impacted by language models is a difficult feat. However, determining appropriate sentiment positions for large groups of people risks marginalizing minority voices. [24] analyze the power hierarchies among groups developing AI policies in a global context, demonstrating the need to include marginalized voices in the policy development process. [26] describe the need for datasets to be carefully collected in their original context so they are not only representative, but also respect and behave appropriately toward those from whom we collect data. These practices must be installed in sourcing PALMS datasets.

In order to update model behavior to what is culturally appropriate and safe, AI researchers must collaborate across fields and sectors to understand what constitutes appropriate and safe sentiment and by what lens. We encourage technical and social sciences to work with policymakers and community representatives across all groups affected by AI systems to build safer, more inclusive systems.

## 6  Questions for Further Exploration

While the *values-targeted dataset* we crafted was for research purposes, adjusting model behavior to be minimally harmful in a given social context requires determining what is appropriate behavior. These experiments sparked the questions for the research community around accountability, scaling laws, generalizability, and other generative models. See Appendix N for questions.

# 7 Limitations

This research was only conducted in the U.S. English language and analyzed through limited evaluations that provide a small window into the models. Since PALMS may be extrapolating from the model's pretraining set, this process may have difficulty replicating underrepresented cultural contexts. We also only evaluated test prompts with a question-answer format and with leading open-ended prompts for gender, religion, and race. Our prompts have varying levels of specificity, as prompt-creation is extremely resource-intensive. Due to the general-purpose nature of language models, these formats may not generalize to all possible downstream tasks. We encourage additional prompts and formats for those applying PALMS to their models. Evaluating alignment and harmful outputs cannot be done by any one metric and means of evaluation is a constantly growing field of research. Quantitative evaluations especially are meant to compare models along the specific axis being measured and are not a comprehensive means of evaluating racial or any other bias. Additionally, working with human evaluators introduces varied perspectives on a difficult task.

# 8 Discussion

The toxicity graphs show that PALMS significantly improves language model toxicity. According to our probes, *base models* consistently scored higher toxicity than our *values-targeted models*. We found that the similarly high-quality writing that our *control models* were fine-tuned on did not give similarly low toxicity results as our *values-targeted models*. Although derived from high-quality text, such as encyclopedic passages, our controls were not designed to reflect any given behavior and so did not show high performance in our evaluations. The quality and the sentiment in the *values-targeted dataset* is critical to ensure the resulting model exhibits desirable behavior.

The Human Evaluations show that PALMS significantly improves *values-targeted model* ratings on the specific axes that we selected, with the largest improvements in the largest models. A scaling law could be drawn from these results: as language models continue to grow in size exponentially, linearly fewer examples should be needed to produce the same major behavioral changes.

In our co-occurrence evaluations, our *values-targeted models* broadly showed more neutral sentiment in top descriptive words across gender, religion, and race than both the *base models* and *control models*.

Of our test prompts, 34 out of 40 did not have similar prompts in our *values-targeted dataset*. Similar prompts include "What would make me more likable?" from our *values-targeted dataset* and "How do I make my classmates like me?" from our test set. Most topics in probes across categories were not covered in the *values-targeted dataset*. The overall high performance in the Human Evaluations suggests that the model is generalizing from the topics and behaviors covered in the *values-targeted dataset*. We speculate the model is extrapolating from similar text in GPT-3's pretraining set that exhibits the desirable behavior embodied in our *values-targeted dataset*. Given the breadth of GPT-3's pretraining data, nearly any position could theoretically be supported in model behavior through PALMS.

# 9 Conclusions

The social contexts in which a language model is developed and deployed play an important role in outlining values for alignment and determining and mitigating harmful outputs. We take this into account when crafting a *values-targeted model* that performs well across the topics we probe according to the positions we outlined for desirable behavior.

We found that fine-tuning on a small but curated dataset can help improve language model behavior and have a larger impact as model size increases. We were surprised we could make so much progress on alignment with a dataset this small. This implies that significantly adjusting the behavior of a large language model is feasible with a small dataset, and human input and oversight is feasible in this method of model value alignment.

## Acknowledgments and Disclosure of Funding

Thank you to OpenAI for funding this project in its entirety.

Thank you to the entire OpenAI technical team for developing and maintaining large language models like GPT-3 that allowed us to conduct our experiments.

Thank you to Sandhini Agarwal and Melanie Subbiah for developing the co-occurrence evaluation suite for GPT-3.

Thank you to John Schulman for developing the tool we used for fine-tuning the models we used in our experiments.

Thank you to Surge.AI[31] for conducting the human evaluations using their human rating service.

Thank you to Alec Radford, Joshua Achiam, Gretchen Krueger, Steve Dowling, Jan Leike, Lilian Weng, Miles Brundage, Greg Brockman, Mira Murati, Timnit Gebru, Iason Gabriel, Yonadav Shavit, Maarten Sap, Boris Power, Jeff Wu, Ryan Lowe, Elizabeth Barnes, Arvind Neelakantan, Long Ouyang, Peter Welinder, Cullen O'Keefe, and anonymous reviewers for their feedback on our earlier versions of this paper. Thank you to Katie Mayer, Luke Miller, and Jack Clark for their help in planning this project.

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
