# OpenReview forum: "Process for Adapting Language Models to Society (PALMS) with Values-Targeted Datasets"
_NeurIPS.cc/2021/Conference — NeurIPS 2021 Spotlight_

### Official Review · Reviewer_gwYe · 2021-07-09

**Rating:** 6
**Confidence:** 4

**Summary:**

Language models exhibit harmful biases. The authors try to address the toxic behaviors of the GPT-3 model as follows. First, they construct a set of examples that correspond to a large set of desired “values” (e.g., gender equality, realistic beauty standards). The examples are prompt-continuation pairs. Then, they finetune the language model to generate the continuation (which has the desired values) given the prompt.

The authors find that using a very small set of examples, the post-finetuning models would generate continuations that are less toxic, using both quantitative (human-eval scores and automatic toxicity scores) and qualitative evaluation methods that analyze word associations given a specific gender, religion, or race.

The paper targets U.S. values only which are not necessarily the universal values; however, the authors acknowledge and point this issue out, in the introduction and the discussion sections.


**Limitations And Societal Impact:**

The project is done using American English, and the project caters to a specific set of U.S. values. However, the authors acknowledge the issue and claim that a similar procedure to fix a biased language model can be done in the context of other cultures as well.

IF the post-finetuning models generalize to downstream tasks (i.e., the tasks still achieve the same performance; there is less bias in downstream tasks), then this work would be encouraging for the NLP/ethics community because it claims that using a small set of expert-written examples could greatly debias a language model.



**Main Review:**

Issue on formatting: The NeurIPS website says “you must format your submission using the NeurIPS 2021 LaTeX style file which includes a preprint option for non-anonymous preprints posted online” but the font of the submitted article does not follow the provided tex file.

(+) The sensitive topics are chosen according to official sources like the government and the United Nations guidelines. They cover a wide range of categories.

(+) Significant testing of the quantitative results (when comparing the post-finetuning model and the pre-finetuning model) is comprehensive.

Lots of critical information, which is required to judge the quality of the paper, is hidden in the 30-page-long appendix; e.g., the methodology for choosing the number of prompts, how specifically the authors wrote the prompts. However, I appreciate the detailed explanations of the prompts in the appendix.

(-) Hyperparameter tuning seems arbitrary (e.g., “training loss weight was 0.1 for the prompt and 1.0 for the completion”, “using learning rate 30x less than the default learning rate” in line 1325). Training details should be expanded. Important parts can be moved to the main text. It’s unclear how long the finetuning lasts. What are the criteria for “convergence”?

(-) Currently the train/dev/test prompts are similar in terms of style. The models can generate better continuations when using top-p sampling (p=0.8) & when there is a prompt that follows a specific format (e.g., one sentence long, usually under 15 words, ending in question marks, etc.). Given the evaluation strategy, it’s unclear whether the improvement in values can transfer to downstream applications.

(-) It’s unclear whether the post-finetuning models still do as well as the pre-finetuning models on GLUE/SuperGLUE classic benchmark tasks.

(-) Missing citations: Nangia et al. (EMNLP 2020), https://arxiv.org/pdf/2010.00133.pdf, which deals with the discussion on categories of biases as well as the evaluation of LM bias. A comprehensive discussion on previous LM-bias-related works should be included; e.g., http://web.cs.ucla.edu/~kwchang/publications_area/.

I would hope for more information related to human factors.
- Who are the writers? How are the writers trained?
- How do the authors validate the continuations?
- Who are the human annotators? How are they trained?
- How do the authors validate the annotation results?

Minor:
- Footnote 9: "top-p controls diversity via nucleus sampling" but top-p sampling and nucleus sampling are the same thing.
- A space before bracket citations?

**Time Spent Reviewing:**

3-4

---

> ### Comment · Area_Chair_P6Qf · 2021-08-16
> **Style guide point**
>
> I agree that it's odd not to use the standard font, but many conferences do allow Computer Modern as an option for some reason, and from a quick skim of the style guide for NeurIPS, I see this ambiguous language that arguably allows for alternative fonts at the same size:
>
> "Times New Roman is the preferred typeface throughout"
>
> Do you see a clearer rule that applies here? If not, I can flag this to the PCs.

---

> > ### Comment · Reviewer_hQiK · 2021-08-17
> > **Reactions to concerns**
> >
> > The hyperparameter tuning and the missing citations seem pretty minor in my opinion. I'm also not too concerned about the exact annotation/validation details; I agree these would be nice to include in the paper, but this seems very easy for the authors to fix, and it doesn't seem like a huge deal regardless.
> >
> > That said, I agree that I would like to see performance on downstream tasks too, and would like to see a comparison between post-finetuning and pre-finetuning performance. These seem like moderately important experiments to run, and would make the paper noticeably stronger (especially if performance *does* transfer to downstream tasks, as you mention -- though I don't actually think this is necessary for the current results to be interesting/important). But on balance I don't think these limitations should be dealbreakers for the paper's acceptance.

---

> > > ### Author Response · Authors · 2021-08-20
> > > **Response to Hyperparameter Fine-tuning**
> > >
> > > Thanks for the response! The hyperparameter tuning is reflective of internal experimentation that was out of scope for this paper. However, future publications from our organization are likely to release these details.
> > >
> > > Please take a look at Appendix D: Capability Evaluation Results if you haven’t already, which compares many of the core benchmarks of the base model to our fine-tuned model. Although more benchmarks are out of scope for this paper at this time, please let us know which ones (besides SuperGLUE/GLUE) that you would have liked to have seen.

---

> > > > ### Comment · Reviewer_gwYe · 2021-08-21
> > > > **Thank you authors**
> > > >
> > > > Thanks for the response! Yes I agree that hyperparameter tuning details are not grounds for rejection.

---

> > ### Comment · Reviewer_gwYe · 2021-08-18
> > **Alternative fonts are fine**
> >
> > Yes I agree with you. I clicked into the provided style file and saw "Times New Roman is the preferred typeface throughout" so it seems okay to change the font. Thanks for catching this.

---

> ### Comment · Reviewer_gwYe · 2021-08-18
> **The only potential dealbreaker imo**
>
> I mostly agree with reviewer `hQik`. Here is the only potential dealbreaker.
>
> Currently the models can generate good continuations (with less toxic content) if we provide a prompt that's similar in style (i.e., one sentence long, usually under 15 words, ending in question marks, etc.). If we use this updated language model in downstream tasks, will the improvement transfer to downstream tasks?
>
> Intuitively given how "powerful" GPT-3 is, possibly/maybe? But this is just an intuition. Moreover when this work gets published, there's a chance that we'll see misleading news headlines (an extreme example could be "NLP models are less toxic now!").
>
>
> Nevertheless, I'm impressed by the quality of prompts described in the appendix. I think the categories and prompts are very useful to future research, so I'm improving my score to 6.

---

> > ### Comment · Reviewer_gwYe · 2021-08-21
> > **Changed my mind again; leaning rejection**
> >
> > Apologies. After more deliberation, I'm reducing the score to 5 again.
> >
> > Although the categories and prompts are very useful and the research topic is extremely important, there's just no strong experimental evidence to support the fact that the language model will reduce bias on downstream applications. It might not even work if the prompts have a different format.
> >
> > To be fair, the paper is not claiming that the revised GPT-3 will do better on downstream applications. The paper is not claiming that the prompt-relatd robustness is good, either. But I'm just afraid that some readers (especially non-academics) might have the wrong takeaway. If the paper is accepted, there needs to be a serious discussion on the above issue.

---

> > > ### Comment · Reviewer_LP2G · 2021-08-21
> > > **What sort of downstream tasks are you thinking of?**
> > >
> > > To clarify: what sort of downstream tasks are you thinking of? Maybe even spelling out the concrete experiment that you want to see could be useful. I'm personally of the opinion that performance on downstream tasks is nice to have but not necessary, though I want to make sure we're on the same page wrt what a "downstream task" is before I reply in more detail.

---

> > > > ### Comment · Reviewer_gwYe · 2021-08-21
> > > > **One example**
> > > >
> > > > Currently, the prompt-continuation has the question-answering format. As I discussed in the review, the style of the questions is similar.
> > > >
> > > > When deploying the model to the real world, the prompt might not be a question or they might not be explicit. There might not be prompts at all. Say, if we use the proposed version of GPT-3 in a chatbot, will it reduce toxicity?
> > > >
> > > > For example, using the metrics in this (https://arxiv.org/pdf/2010.00133.pdf) paper and their dataset (see table 1 and section 3), I'm not convinced that the proposed GPT-3 model will do better.
> > > >
> > > > Again, the paper wasn't overclaiming, but I'm just afraid that the general public will have the wrong takeaway, without more discussion.

---

> > > > > ### Comment · Reviewer_LP2G · 2021-08-21
> > > > > **On Overclaiming**
> > > > >
> > > > > Got it, thanks for clarifying. FWIW, I do feel like the paper does overclaim in certain settings---for example, L41, where they cite "applications" as a motivating factor:
> > > > >
> > > > > ```
> > > > > Given how desirable behavior for a language model may differ by application, training a language model from scratch for each application’s desirable behavior is not scalable.
> > > > > ```
> > > > >
> > > > > Given this, I think that your issue is that their evaluation / method does not seem particularly grounded in any such application / downstream task, and that the method could be difficult to apply to different applications / things might not transfer across applications (let me know if I'm incorrect here). I agree that this is a concern that the paper doesn't really address empirically, and the authors should consider adding some more discussion about this.
> > > > >
> > > > > However, I agree with the authors in this [comment](https://openreview.net/forum?id=k-ghaB9VZBw&noteId=qfOHA_OHuOL) that, from reading the rest of the paper beyond that one snippet, I actually felt like they were studying adaptation to cultural contexts, and that this is a first step in that direction. As a result, I'm not sure whether your dissatisfaction stems from (1) overclaiming (i.e., they purport to target downstream tasks, so they should have experiments that present results on downstream tasks) or (2) substance (i.e., targeting downstream tasks is important, and the lack of such results means that the paper does not have enough substance for a NeurIPS paper).
> > > > >
> > > > > If (1), I feel like this could be addressed by the authors simply rewriting a bit and clarifying / tempering their claims? This seems like a very fixable issue that doesn't necessarily need to go through another round of review (though, frankly, I feel like overclaiming in titles and introductions has become more of the norm than the exception). If (2), I think we should agree to disagree---despite the limited and slightly-toy setting, I still think the work is interesting, informative, and useful, and thus think that it'd be better published than not.

---

> > > > > > ### Comment · Reviewer_gwYe · 2021-08-22
> > > > > > **Agreeing with your comments reviewer LP2G**
> > > > > >
> > > > > > Thank you for the insightful discussion, reviewer LP2G and the authors!
> > > > > >
> > > > > > Yes, I do agree that...
> > > > > > - More discussion on the limitation of the work is warranted. For example, the authors should discuss the settings where the prompts are not too explicit, or when there are no prompts. Section 3.4 is in the right direction but I think the prompts in the section is still too explicit.
> > > > > > - The work is interesting, important, and as you said it's a "first step in that direction" of adapting to cultural contexts. Raising score to 6 again.

---

> > > > > > > ### Author Response · Authors · 2021-08-23
> > > > > > > **Thank you, Reviewers, and Will Update Paper**
> > > > > > >
> > > > > > > Thank you reviewers LP2G and gwYe for your in-depth discussion and feedback! We know you both have put a lot of time into this and we appreciate your hard work making this submission better. You both raise excellent points and we are happy to temper our claims and add your points on downstream applications to the limitations section. We are very happy that you both agree that the work is important to publish, especially on the importance of cultural context to bias.

---

> > > ### Author Response · Authors · 2021-08-21
> > > **Overall goal & other formats presented**
> > >
> > > Thank you for your response and we’re glad you agree it’s an important topic! We aren’t aiming to overall reduce bias in downstream tasks, but to provide a novel framework and process to adapt language models to cultural contexts. As we assert bias is normative, we test according to the set of values we established. This limits evaluations we can run.
> > >
> > > We encourage you to review Section 4.2 and Appendix F that describe the qualitative bias probes we conducted, which were all in a different format (format is described in Section 4.2). We believe that these probes demonstrate promise for good performance on downstream applications.

---

### Official Review · Reviewer_hQiK · 2021-07-16

**Rating:** 7
**Confidence:** 4

**Summary:**

This paper proposes an approach to aligning language models with target values. The authors construct a small hand-crafted dataset that reflects a set of intended values, then finetune large language models on this dataset. They find that this simple approach significantly improves performance according to a few different metrics.

**Limitations And Societal Impact:**

This paper is largely focused on addressing a problem that is societally important. I think it mostly does a good job of this. One point I would probably want it to emphasize somewhat more is that the target values to finetune on are not only subject to some debate, but may also vary from application to application even if there is consensus (e.g. because it may depend on whether a model is making descriptive claims or normative claims). But especially given that they're mainly proposing a general framework/approach rather than, say, a particular dataset to finetune on, this is a pretty minor point.

**Main Review:**

This paper addresses an important problem (controlling the outputs of language models, especially in ways that are more aligned with human values). It proposes a simple method for addressing this problem that indeed seems to improve performance significantly. The experiments generally seem convincing, and the improvement seems reasonably large.

The approach taken by the paper is *extremely* simple and essentially amounts to finetuning (albeit on a dataset constructed in a particular careful way). I think some would view this as a drawback, and it does admittedly makes the paper feel a little on the thin side in terms of originality, but for the most part I think this is a positive thing. Indeed, one of the main takeaways is that one can do a good job using this method without much effort by using an extremely small number of examples (in the appendix they claim that performance starts to converge at around only 60 samples, at least for the 175B model).

The paper itself is also well-written, with a very detailed appendix. Overall I think it's relatively straightforward, but well-executed, with a couple arguably surprising results, and makes meaningful progress in both addressing and understanding a very important problem.

Other feedback:
- I think it's an important observation that the approach becomes more effective with model size, but to the extent that larger models are more sample efficient this isn't too surprising. I'd like to know how much of this is due to larger models converging to a higher level of performance vs merely being more sample efficient than smaller models. To be fair, they do say on the last page that future work should assess scaling laws for this problem, which is basically what I'm advocating here. But it seems especially important in this context because it affects how scalable this approach will be moving forward. I'd be less excited about it if peak performance is pretty much the same but converges at 6 samples instead of 60 samples with GPT-4 or whatever.
- I'm also curious to know how finetuning on this hand-crafted dataset affects performance out-of-distribution. For example, if you have a rather different input format (such as using the 175B model as a chat bot having a long back-and-forth conversation with a human), does it still tend to output more aligned outputs?

I think it would be quite a bit stronger with these sorts of additional experiments, which is why I'm only giving it a 7, but I think it still probably deserves to be accepted.

Minor comments;
- ALMBFVD is a very cumbersome name, so I would seriously consider changing it. Even just FVD would be much better in my opinion.
- Many of the figures could be much nicer (partly aesthetically, but especially in terms of ease of understanding what's going on).

**Time Spent Reviewing:**

3

---

### Official Review · Reviewer_LP2G · 2021-07-17

**Rating:** 8
**Confidence:** 4

**Summary:**

This paper studies the feasibility of using a small, carefully-constructed corpus to guide large language modeling toward particular values embodied by the corpus. In particular, the authors perform a case study of whether such a process can make GPT-3 more sensitive, according to a Western, US American lens.

The authors describe an iterative procedure for collecting a dataset that reflects the targeted values and evaluating whether the language model has improved, according to the targeted standards. They apply their procedure to several topics and language models of varying sizes, showing that the process is generally successful for steering language models toward the desired values.


**Limitations And Societal Impact:**

Yes, I found the discussion to be satisfactory, and this paper in particular is one that requires such a discussion.


**Main Review:**

Originality: This paper is quite original, and is the first of my knowledge to contribute such an analysis for language models of the scale they study.

Clarity: I thought this paper was clear. I appreciated the authors’ efforts toward contextualizing the lenses / values that they are coming from, and recognizing that different people and different cultures have different lenses. The description of the data collection process and evaluation was also pretty easy to follow.

Quality / Significance: I thought that this paper was reasonably thorough in controlling for potential confounds. I wish there was a bit more discussion about why high-quality data in general (the “control” text) decreases language model toxicity---do the authors think that this is specific to the particular values they target? I also appreciated the level of detail to which the authors described the data collection and evaluation process. I wish there was more space in the main paper to highlight results across different topics, but the paper is quite content-full as is.

There are clearly many open questions left: how well does this method work for values that aren’t as widely held? Perhaps this method only works well if the language model has seen text that is broadly representative of the target values during pretraining. How can we have more fine-grained control over what prejudices are replaced with (e.g., the top descriptive word for “Muslim” switching from a bias toward Islamism to a bias toward heterosexuality).

Despite these questions, I think that this study will be of interest to the broader community, and that the work is significant.


**Time Spent Reviewing:**

2

---

### Decision · Program_Chairs · 2021-09-27

**Decision:**

Accept (Spotlight)

**Comment:**

This paper demonstrates that large language models (i.e., GPT-3) can be relatively cheaply fine-tuned so as to generate text continuations consistent with some specific values/toxicity-related goal. Reviewers had some concerns about how broadly these results would apply, but nonetheless all found the results to be significant and timely.